# Evaluation of Efflux-Mediated Resistance and Biofilm formation in Virulent *Pseudomonas aeruginosa* Associated with Healthcare Infections

**DOI:** 10.3390/antibiotics12030626

**Published:** 2023-03-22

**Authors:** Paul A. Akinduti, Onome W. George, Hannah U. Ohore, Olusegun E. Ariyo, Samuel T. Popoola, Adenike I. Adeleye, Kazeem S. Akinwande, Jacob O. Popoola, Solomon O. Rotimi, Fredrick O. Olufemi, Conrad A. Omonhinmin, Grace I. Olasehinde

**Affiliations:** 1Microbiology Unit, Department of Biological Sciences, Covenant University, PMB 1023, Ota 112104, Ogun State, Nigeria; gwise60@gmail.com (O.W.G.); hanah.ohore@gmail.com (H.U.O.); samuel.popola@cu.edu.ng (S.T.P.); grace.olasehide@covenantuniversity.edu.ng (G.I.O.); 2Obasanjo Holdings Limited, Abeokuta 110124, Ogun State, Nigeria; ariyo_segu@yahoo.com; 3Veterinary Teaching Hospital, Federal University of Agriculture, Abeokuta 110124, Ogun State, Nigeria; adeleyea@funaab.edu.ng; 4Department of Chemical Pathology and Immunology, Federal Medical Centre, Abeokuta 110124, Ogun State, Nigeria; kakinwande@gmail.com; 5Applied Biology and Biotechnology Unit, Department of Biological Sciences, Covenant University, PMB 1023, Ota 112104, Ogun State, Nigeria; jacob.popola@covenantuniversity.edu.ng (J.O.P.); conrad.omohinmin@covenantuniversity.edu.ng (C.A.O.); 6Department of Biochemistry and Molecular Biology, Covenant University, PMB 1023, Ota 112104, Ogun State, Nigeria; solomo.rotimi@covenantuniversity.edu.ng; 7Department of Veterinary Microbiology and Virology, College of Veterinary Medicine, Federal University of Agriculture, Abeokuta 110124, Ogun State, Nigeria; folufemi@funaab.edu.ng

**Keywords:** *Pseudomonas aeruginosa*, antibiotic resistance, biofilm, efflux pump, virulence factors

## Abstract

*Pseudomonas aeruginosa* is a significant pathogen identified with healthcare-associated infections. The present study evaluates the role of biofilm and efflux pump activities in influencing high-level resistance in virulent *P. aeruginosa* strains in clinical infection. Phenotypic resistance in biotyped *Pseudomonas aeruginosa* (*n* = 147) from diagnosed disease conditions was classified based on multiple antibiotic resistance (MAR) indices and analysed with logistic regression for risk factors. Efflux pump activity, biofilm formation, and virulence factors were analysed for optimal association in *Pseudomonas* infection using receiver operation characteristics (ROC). Age-specificity (OR [CI] = 0.986 [0.946–1.027]), gender (OR [CI] = 1.44 [0.211–9.827]) and infection sources (OR [CI] = 0.860 [0.438–1.688]) were risk variables for multidrug resistance (MDR)-*P. aeruginosa* infection (*p* < 0.05). Biofilm formers caused 48.2% and 18.5% otorrhea and wound infections (95% CI = 0.820–1.032; *p* = 0.001) respectively and more than 30% multidrug resistance (MDR) strains demonstrated high-level efflux pump activity (95% CI = 0.762–1.016; *p* = 0.001), protease (95% CI = 0.112–0.480; *p* = 0.003), lipase (95% CI = 0.143–0.523; *p* = 0.001), and hemolysin (95% CI = 1.109–1.780; *p* = 0.001). Resistance relatedness of more than 80% and 60% to cell wall biosynthesis inhibitors (ceftazidime, ceffproxil, augumentin, ampicillin) and, DNA translational and transcriptional inhibitors (gentamicin, ciprofloxacin, ofloxacin, nitrofurantoin) were observed (*p* < 0.05). Strong efflux correlation (r = 0.85, *p* = 0.034) with MDR strains, with high predictive performances in efflux pump activity (ROC-AUC 0.78), biofilm formation (ROC-AUC 0.520), and virulence hierarchical-clustering. Combine activities of the expressed efflux pump and biofilm formation in MDR-*P. aeruginosa* pose risk to clinical management and infection control.

## 1. Introduction

*Pseudomonas aeruginosa* is one of the significant pathogens identified with increasing healthcare-associated infections (HAIs) [1,2]. It is a ubiquitous and non-fastidious organism that grows in humid or wet environments. This opportunistic pathogen is commonly found among immune-compromised patients, cases of underlying diseases (diabetes), traumatized or invasive surgical procedures, sepsis, bloodstream, otorrhea, catheterized or indwelling devices, skin, and soft tissue infections (mostly related to burns, and pressure ulcer) [3,4,5]. Frontline antibiotics are usually selected to treat *Pseudomonas* infection to prevent growing resistance. Antibiotic selective pressures facilitate intrinsic resistance mechanisms to the drug of choice, making therapeutic applications difficult, expensive, and exacerbating morbidity [6].

*P. aeruginosa* biofilm functionally creates favourable medium that prevents accessibility of antibiotics to infection sites and promotes cell-surface interactions for tissue degradation. Biofilm formation in *P. aeruginosa* infection further enhances in vivo colonization, adaptation and persistence, through aggregation of exopolysaccharides (EPS) and biomolecules (lipids, proteins, carbohydrates) induced by poly-N-acetylglucosamine [7]. The activities of the multi-drug efflux system and low outer membrane permeability are important intrinsic components which are characterized with broad substrate specificity and genetic domains mediating unrelated multi-drug resistance (MDR) in *P. aeruginosa* [8]. MexAB-OprM is one of the major efflux pumps belonging to the resistance-nodulation-division (RND) which plays significant roles in intrinsic and acquired resistance to cell wall biosynthesis inhibitors (such as β-lactam classes), DNA and protein synthesis inhibitors (mostly fluoroquinolones and aminoglycosides respectively) [9,10]. Persistence expression of MexAB-OprM efflux pump of the RND superfamily constituting an inner membrane (MexB), periplasmic membrane fusion protein (MexA) and a channel-forming outer membrane protein (OprM) [10], are preferential and active extruding mechanisms mediating poor permeability, facilitating high-level *P. aeruginosa* resistance and increase virulence factor productions.

Increase efflux activity and virulence factors usually intensify disease morbidity, with activation of hemolysin that cause tissue damage, dysfunctional cellular immune responses, and red cell destruction [11,12]. Involvement of extracellular secretions of protease and lipases (lipolytic enzymes) in biofilm formers aids the functional capacity of the pathogen for invasion of host cells, and subversion of host defenses leading to tissue damage [13]. The ability of the virulence factor to degrade mucosal lipids and phospholipids on the epithelial surface alters mucosal integrity and promotes constant invasion [14]. Combine activities of virulent proteins promote the formation of membrane blebs and lipid rafts, aiding *P. aeruginosa* intracellular survival and cytosol lipid degradation leading to severe tissue pathology [15]. Several reports of antibiotic resistance among the clinical *P. aeruginosa* isolates are associated with biofilm and efflux pump activities [16,17,18,19,20,21]. However, there is paucity of relevant information on the assessment and synergy of efflux pump, biofilm formation, and virulence factors activities with the antibiotic resistance profile of healthcare-associated *P. aeruginosa* and possible implications on antibiotic stewardship in Nigerian hospital settings. The association of biofilm formation, efflux pump activities, and virulence proteins from *P. aeruginosa strains* resulting in high-level multi-drug resistance in clinical infection is yet to be defined. The study aims to evaluate the functional role of biofilm-forming capacity and efflux pump activities of virulent MDR-*P. aeruginosa* isolates from clinical infections and association of these factors with pathogen tissue tropism.

## 2. Results

### 2.1. MDR Pseudomonas aeruginosa Resistance Pattern

Of the 147 *P. aeruginosa* strains collected over the 6-month period, 27 were multi-drug resistance (MDR), showing significant resistance of more than 80% to cell wall biosynthesis inhibitors (ceftazidime, augmentin, ampicillin) and over 60% resistance to DNA translational and transcriptional inhibitors (ofloxacin, ciprofloxacin, nitrofurantoin); (*p* < 0.05, Figure 1A). To identify antibiotic resistance relatedness of the MDR *P. aeruginosa*, clustergram was built with strains from clinical samples. The heatmap revealed a comparative and related resistance pattern to cefprozil, ofloxacin and gentamicin (Figure 1B). Overall low median susceptibility rate of less than 10% was shown by the MDR *P. aeruginosa* obtained from all the clinical samples (Figure 1C).

### 2.2. Risk Factors for MDR Pseudomonas aeruginosa Infection

The univariate analysis of the strain collections from the subjects presenting various systemic infections is shown in Table 1. Age-specificity of the subjects showed a significant risk for MDR *P. aeruginosa* infection (OR = 0.986, CI = 0.946–1.027, *p* = 0.024) with the highest prevalence rate among children ages 0–12 years (47.6%) and adult ages 24–50 (25.7%). Higher rate was recorded among female (20.8%) compared to male (OR = 1.440, CI = 0.211–9.827, *p* = 0.001). Considerable high risk for otorrhea (37.0%), wounds (21.4%) and bloodstream (18.2%) infections were noted (OR = 0.860, CI = 0.438–1.688, *p* = 0.002), but no MDR *P. aeruginosa* strain was isolated from other infection sites (including the eyes, throat, lungs and endocervical).

### 2.3. Phenotypic Virulence Factor Expression

Of the MDR*-P. aeruginosa* strains (Appendix A), significant rates of 48.2% and 18.5% (95% CI = 0.820–1.032; *p* = 0.001) were biofilm producers causing otorrhea and wound infections respectively. Strains found in otorrhea infection expressed significant efflux pump activity (44.4%; 95% CI = 0.762–1.016; *p* = 0.001), protease (18.5%; 95% CI = 0.112–0.480; *p* = 0.003), lipase (40.7%; 95% CI = 0.143–0.523; *p* = 0.001), and hemolysin (33.3%; 95% CI = 1.109–1.780; *p* = 0.001) compared to strains from other infections (Table 2).

### 2.4. Correlation and Prediction Performance of EP Activity and Expressed Biofilm

Assessment of correlation coefficients of biofilm, efflux pump activity and virulence factors with MDR *P. aeruginosa* was shown in Figure 2A,B. A strong correlation of the efflux activity (r = 0.85, *p* = 0.034) with the MDR strain mostly from the ear infection, was recorded. Higher proportion of MDR strains overproducing efflux and biofilm clustered together (cluster a) compared to less number recorded in cluster b (Figure 2A). In clusters c and d, less proportion of strains (producing protease, lipase and hemolysin) from ear, wound, urine, vaginal and bloodstream infections further confirm low correlation as shown in Figure 2B. Figure 2C showed the Receiver Operating Characteristic (ROC) curves for predicting efflux and biofilm production with MAR indices of MDR *P. aeruginosa*. High biofilm ROC-AUC 0.78 (95% CI: 0.580–0.915) highlights a stronger predictive performance compare to efflux ROC-AUC 0.520 (95% CI: 0.321–0.715), showing a significant phenotypic adaptive resistance mechanism (Figure 2C). Weak MARI correlation with protease, lipase and hemolysin was noted based on the strain frequency from the clinical samples and biofilm production (Table 3). Hierarchical-clustering heat-map provided three clusters of MDR *P. aeruginosa* compared to age, sex and efflux proteins (MexA and MexB) with the degree of biofilm, hemolysin and lipase production (Figure 3). Four main clusters defined the degree of strain resistance relatedness. Cluster b included strains with high MexA and MexB encoded genes, hemolytic and lipase productions, and a mild MARI level, while strains in cluster c expressed high-level MARI and biofilm production, mostly among females aged 0 to 40 years.

## 3. Discussion

In the current study, age–specificity is a considerable risk factor for MDR *P. aeruginosa* infection rate showing a significant prevalence among children. Low immunity due to poor nutrition and maternal care, persistent nosocomial infection during admission, and consistent use of medical devices (urinary catheters, intravascular and endotracheal tubes etc), facilitate strain colonization and high infection rate [22,23]. The declining demography of the admitted adult group in these healthcare settings with the poor socio-economic condition, prolonged hospital stay, and antibiotic misuse are common factors that persistently influence the infection spread [24]. Self-prescription and consumption of counter drugs are common practices that enhance MDR and continuous treatment failure. The high rate of recorded MDR *P. aeruginosa* among female patients presents a different *Pseudomonas* infection phenomenon attributed to hormonal changes during the menstrual cycle, influencing behavioral and physiological changes that alter the immune responses [25]. Increased poor clinical outcomes in several gynecological conditions and urinary cases are potential risks for *Pseudomonas* infection [26]. Though ear infection (otorrhea) is observed to be frequent among infants and young children [1], poor clinical management by quacks, unregulated antibiotic prescription, and unhygienic perforation of the ear lobe usually facilitate the middle ear infection. Observed *Pseudomonas* wound infections are exacerbated by expired or ineffective topical antibiotics, contaminated or poorly sterilized dressing gauze. These further pose a significant challenge to clinical management, causing prolonged hospital stay and increasing morbidity [9]. 

Furthermore, the produced biofilm matrix plays structural function of inhibiting neutrophil elastase (a bactericidal enzyme) from the host to enable persistence infectivity and virulence [21]. The ability to produce protease once attached or colonized with biofilm intensifies necrotization of epithelia mucosa and functional invasive mechanism. As *P. aeruginosa* protease degrades basement membranes and extracellular matrix components, it cleaves the laminin, fibronectin, proteoglycans, and various collagens, causing direct tissue invasion with extensive ulceration leading to the down-regulation of cascade path of humoral immune responses [27]. During invasion, *P. aeruginosa* lipase (an extracellular lipolytic enzymes), interacts with inflammatory mediators released from human platelets, neutrophilic and basophilic granulocytes, and monocytes to enhance massive tissue damage and bloodstream infection [28]. Hemolysin production further suggests impending disease severity by forming cellular pore and necrotic cell death. As host cell lysis increases, there is increase in erythrocytes depletion, cytotoxicity, host cell plasma membrane fluidity, bilayer structure deformities resulting to immune evasion, physiological imbalance and septic shock mostly in severe conditions [29]. These virulence factors contribute to several stages of tissue damage that involve adherence, colonization and invasion leading to local and systemic dissemination of biofilm lipopolysaccharide, polysaccharide slime (alginate), hemolysin, and proteases [29].

Clustering of highly related MDR *P. aeruginosa* obtained from wounds, urinary tract and ear infections reveal active circulation and re-dissemination of strains with potential to exchange resistant genetic elements (plasmids, transposons, integrons and prophages), thereby facilitating horizontal gene transfer. This results to relatively high recombination that could give rise to multi-clonal MDR population [30]. Dissemination of these MDR strains portrays an imminent health risk that could escalate to an outbreak of multiple extra-intestinal infections but the inclusion of genomic-wide surveillance centered on the community antimicrobial stewardship is highly important to curb the spread. Low ciprofloxacin and ceftazidime susceptibility presented possible intrinsic resistance to the available drug of choice [31]. Hospital and community *Pseudomonas aeruginosa* infection control could suffer major setbacks due to high-level resistance to cell wall biosynthesis inhibitors (particularly augmentin, ceftazidime and ampicillin) which are most prescribed antibiotics. The recorded antibiotic resistance of more than 60% to translational and transcriptional inhibitors poses a serious challenge to clinical management with a remarkable capacity to develop an extend-spectrum resistance to other classes of antibiotics. Low outer membrane permeability, high efflux activity and producted inactivating enzymes are possible intrinsic or acquired resistance gained through horizontal gene transfer or mutational changes [32], limiting antibiotic efficacy. The formation of biofilm is one of the *P. aeruginosa* adaptive resistance mechanisms, acting as a diffusion barrier to limit antibiotic access to the bacterial cells making the strains capable of surviving antibiotic attacks [7]. Hospital and community antibiotic stewardship needs to be strengthened with proper information on combining these antibiotic classes and regulation of drug prescription, particularly in local outlets.

Very low clinical MDR *P. aeruginosa* susceptibility corroborate the previous report of increasing MDR in similar clinical settings in northern and southern Nigerian with more prevalence rate of hospital-acquired *Pseudomonas* infection [18,19,20,33], which is evident in different infection sources. Newer clinical approaches are needed to curtail the increasing resistance by considering the innovative integrated system in prescription and therapeutic formulation, with combined synergistic mechanism of action. 

Over-expression of a well-developed efflux system embedded across the outer membrane (OM) permeability barrier continuously enables active expulsion of antibacterial compounds, improving the cell persistence and multiplication during infection [34]. Reports have shown that *P. aeruginosa* OM carries several substrate-specific porins that limit the intake of certain compounds and nutrients, further enhance expression of MexAB-OprM efflux, which provides resistance to varieties of antibiotics (chloramphenicol, β-lactams, quinolones, macrolides, tetracycline and novobiocin) [35]. The abundance of porin on the OM constitutes high permeability properties that usually influence the active expulsion of accumulated antibiotics [36]. The weak association of the resistance pattern of the isolates with produced biofilm and virulence factors indicates the interdependent strategies of *P. aeruginosa* to intensify infection while maintaining high-level resistance. The interplay of virulence factor activity and biofilm formation demonstrates a dominant influence that increases infection burden, therapeutic failure, morbidity and poor quality of life [36]. The observed association of the biofilm-forming ability, virulence factor, and efflux activities provides insight into the mechanism of *P. aeruginosa*-host adaptation in severe infection. Further genomic studies would be needed to ascertain the pathways that facilitate the MDR-*P. aeruginosa* pathology. Clustering efflux pump and biofilm-producing *P. aeruginosa* obtained from different infection sources with related multi-antibiotic resistance indices reveals a remarkable MDR strains relatedness that could trigger an hospital and community-acquired *P. aeruginosa* infection which may be difficult to treat.

The predictive performance of the biofilm matrix indicates phenotypic adaptations and protective capacity provided by the matrix for cellular efflux pump and persistent induction of modulators (phosphonate degradation, lipid biosynthesis, and polyamine biosynthesis) that contribute to persistent resistance [37]. Functional mediation of biofilm enhances genetic regulatory pathways involving multidrug efflux activities and possible alteration of drug targets [38]. These findings are helpful in understanding biofilm-mediated resistance in *P. aeruginosa* infection pathology and important gene targets to be considered for new drug discovery. Comparative analysis of hierarchical clustering of MDR strains characterized with a combination of efflux genes, virulence factors, and biofilm further suggests a major impending public health challenge that calls for routine surveillance, monitoring, and definitive strategic interventions. The study provides a clue to the emerging clusters of virulent MDR *P. aeruginosa* that could be regarded as high risk for public health.

## 4. Materials and Methods

### 4.1. Bacterial Strain Collections

One hundred and forty-seven *P. aeruginosa* isolates from diagnosed disease conditions were obtained from clinical samples of patients attending two major healthcare facilities; Federal Medical Centre, Abeokuta and General Hospital, Ota; in southwest Nigeria between September 2020 and March 2021 with ethical approval from Covenant Health Research Committees, Covenant University, Ota, Nigeria (CHREC/055/2020). All the isolates were preserved in semi-solid Brain Heart Infusion (BHI) (Oxoid, Basingstoke, UK) supplemented with glycerol and re-characterized for confirmation following standard biochemical methods previously described [8].

### 4.2. Antimicrobial Susceptibility

The susceptibility profile of commonly prescribed antibiotics for *Pseudomonas* infections was determined using Kirby-Bauer disc diffusion in accordance with CLSI recommendations and guidelines [39]. Overnight culture from cetrimide agar was sub-cultured on BHI and further incubated for 24 h at 37 °C. Bacterial suspension of 0.5 MacFarland turbidity was spread on BHI using sterile swab stick. Antibiotic discs including ceffproxil (CPR, 10 µg), ofloxacin (OFL, 30 µg), amoxicillin-clavulanic acid (AUG, 5/30 µg), nitrofurantoin (NIT, 30 µg), ciprofloxacin (CPX, 30 µg), ceftazidime (CAZ, 30 µg), gentamicin (GEN, 10 µg), and ampicillin (AMP, 30 µg) were gently placed on the plate and incubated at 37 °C for 24 h. Phenotypic resistance was interpreted according to the CLSI guidelines (2018) [39,40]. Isolates showing resistance to at least one agent in more than three classes of the antibiotic group were classified as multi-drug resistant *P. aeruginosa* (MDR *P. aeruginosa*) according to Magiorakos et al. [41]. Multiple antibiotic resistance index (MARI) ranging between 0 and 1 was determined by dividing the total number of detected resistance to antimicrobials for each isolate by the total number of tested antimicrobials [42].

### 4.3. Biofilm Assay

The formation of biofilm was assayed in a microtitre plate as described by Mathur et al. [43]. Briefly, 200 µL of 0.5 McFarland inoculum were placed in 96-well microtiter plate along with negative control. After incubation at 37 °C for 24 h, broth suspension was discarded and rinsed three times with sterile water to remove the planktonic bacteria cell. To air-dried plate, 200 µL of 1% (*v*/*v*) crystal violet solution was added and further incubated at room temperature for 1 h. Following two washing to remove the stain, 200 µL glacial acetic was added to dissolve the attached stain. The optical density (OD) was measured using UV Microplate reader [44].

### 4.4. Phenotypic Detection of Strain Efflux Pump (EP) Activity

MDR *P. aeruginosa* strains were selected for EP activity and biofilm detection. The EP activity was assayed with the Ethidium Bromide (EtBr) which is a substrate of efflux pumps using agar cartwheel method described by Ugwuanyi, et al. [21]. Adjusted 0.5 MacFarland turbid broth suspension was streaked on Mueller–Hinton agar plates containing 0 mg/L, 0.5 mg/L, 1 mg/L, 1.5 mg/L, and 2 mg/L concentrations of EtBr and incubated for 24 h at 37 °C. The plates were examined after incubation for fluorescence under UV transilluminator, and isolates that did not fluoresce were identified to express active EP activity and were scored according to the concentration of EtBr. In contrast, bacteria that fluoresced at a minimum concentration of EtBr were recorded as not possessing active efflux pumps.

### 4.5. Phenotypic Detection of Virulence Factors

Hemolysin production was demonstrated as described by Edberg et al. [45]. Overnight single colony was sub-cultured on 5% defibrinated sheep blood agar overlaid on the Nutrient agar base (Oxoid, UK) and incubated at 37 °C for 72 h. A clear halo zone indicating lysis of the red blood cells around the colony indicates haemolysin production [46]. A phenotypic assay for protease production was performed according to Suganthi et al. [47]. Briefly, broth of 0.5MacFarland was dropped on Skim milk agar supplemented with 1% casein and allowed to be adsorbed and incubated at 37 °C for 24 h. Casein hydrolysis was positive, indicating a clear zone around the inoculum spot. A loopful of 18 h bacteria colonies was streaked onto tributyrin agar plate, incubated at 37 °C for 24 h and then observed for a zone of hydrolysis around the colony. The zone of hydrolysis produced by the strain was observed as an indication for lipase production [48].

### 4.6. Efflux Gene Genotyping

Genomic DNA was isolated from pure broth culture using a DNA extraction kit (Zymo, Irvine, CA, USA) according to the manufacturer’s instruction and evaluated for quality and purity with Nanospectrophotometer. Encoded efflux pump genes (*mexA*, *mexB* and *OprM*), which belong to the RND pump, was assayed with PCR [49]. In a total reaction volume of 25 µL, 1 µL of DNA template, 12 µL 5X FIREPOL Solis Biodyne (Tartu, Estonia) master mix, 1 µL each of reverse and forward primers (Appendix A) and 10 µL deionized water (Sigma-Aldrich, St. Louis, MO, USA). DNA template obtained from *P. aeruginosa* ATCC 27853 was used as positive control and nuclease-free water as negative control. The PCR assays were performed in Bio-Rad MJ thermal cycler (T100cycler, Bio-Rad, Hemel Hempstead, UK) with initiation of 95 °C for 5 min; 30 cycles of denaturation at 95 °C for 60 s, annealing (listed in Table 1) for 30 s, extension 72 °C for 45 s and final elongation at 72 °C for 10 min. Obtained amplicons were electrophoresed in 1.5% agarose gel at 100 V for 60 min and visualized under a trans-illuminator.

### 4.7. Data Analysis

Among the patients, risk factors for *P. aeruginosa* infection were analysed with univariate logistic regression to determine the significance of dependable variables taking the odd ratio at 95% confidence intervals (CIs) using SPSS v22. Level of virulence factors production from MDR *P. aeruginosa* were expressed as percentages and further analysed with unpaired *t*-tests (two-tailed, unequal variance) to determine the differences between the variables and *p*-values less than 0.05 were considered statistically significant. The significance of the resistance rate of the selected translational and transcriptional inhibitors was determined with ANOVA in GraphPad Prism 8.0. Median resistance of MDR *P. aeruginosa* obtained from various clinical sample sources was evaluated with ANOVA. The correlation matrix of phenotypic antibiotic resistance with biofilm formation, efflux activity, protease, lipase, and hemolysin productions was calculated using the Pearson method. The possible association of these variables was illustrated in graphical presentations. To predict the intensity of antibiotic resistance of MDR *P. aeruginosa* with biofilm and efflux pump activity, receiver operating characteristics (ROC) curves were calculated, taking the significance of optimal predictions given by the area under the curve (AUC) [50], using the statistical program MedCalc Statistical Sofware (Ostend, Belgium).

## 5. Conclusions

The activities of the produced virulence proteins, efflux pump and biofilm, contribute significantly to high antibiotic resistance and persistent chronic infection with possible transmission and exchange of genetic materials among the MDR *P. aeruginosa* population. Observed association and hierarchical clustering of virulent MDR strains reveal high capacity for effective adaptability, survival, and infectivity. Periodic community-wide intervention, monitoring of local drug prescriptions, and appropriate clinical management of the *Pseudomonas* infections are needed to mitigate the impending spread of virulent MDR *P. aeruginosa*.

## Figures and Tables

**Figure 1 antibiotics-12-00626-f001:**
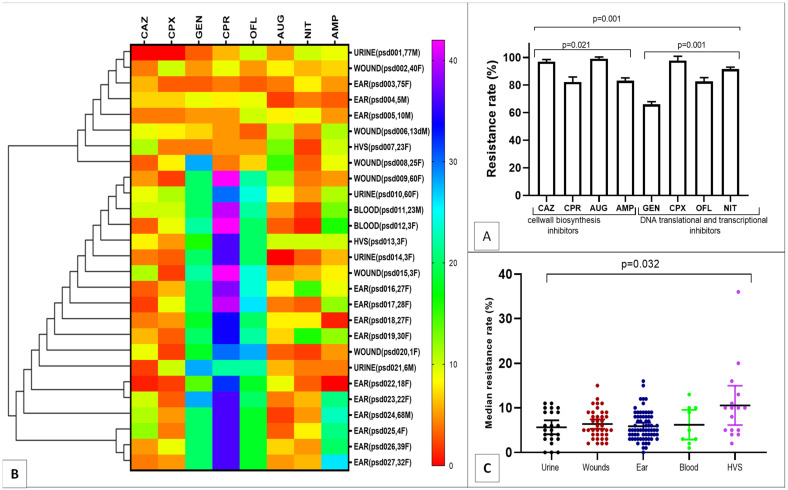
Resistance pattern and Clustergram of the MDR strains ((**A**), Strain resistance to antibiotic classes; (**B**), Resistance relatedness; (**C**), Median resistance rates).

**Figure 2 antibiotics-12-00626-f002:**
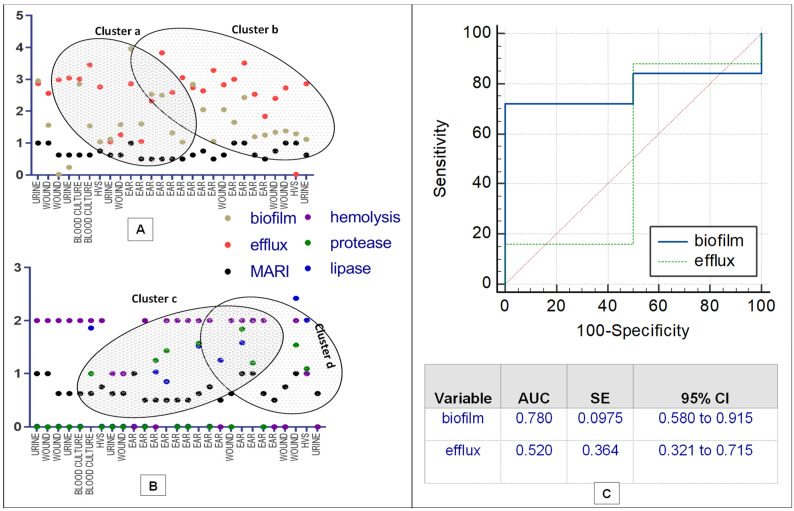
Correlation matrix of phenotypic antibiotic resistance ((**A**,**B**), Association of antibiotic resistance, biofilm formation, efflux activity, protease, lipase and hemolysin production; (**C**), ROC curves).

**Figure 3 antibiotics-12-00626-f003:**
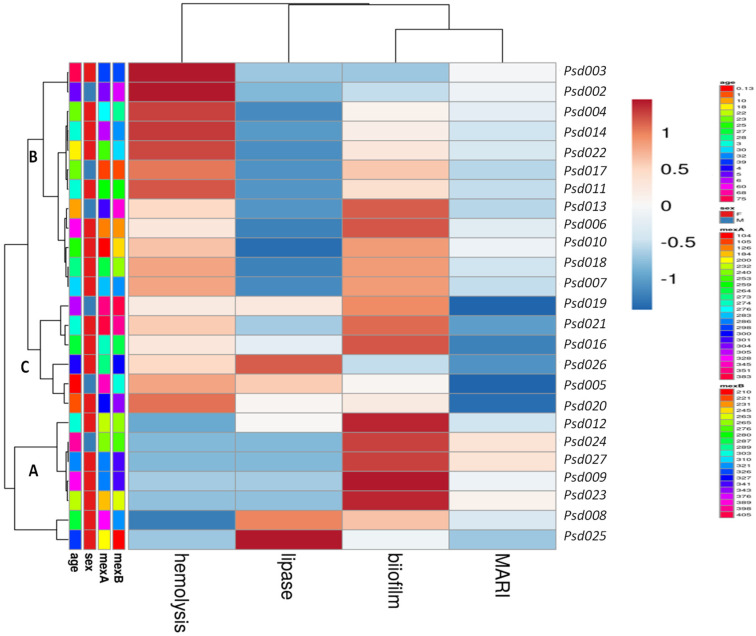
Heatmap of hierarchical clustering of MDR-Psa resistance and virulence profiles in comparison with the age, sex, MexA and MexB efflux proteins (ranging between 158–395 bp), MARI and expressed virulence factors. Rows clustered by UPGMA method using Euclidean distances to define clusters.

**Table 1 antibiotics-12-00626-t001:** Univariate analysis of risk factor for MDR-*P. aeruginosa* infections.

Characteristics	MDR-Psa *n*/N(%)	OR(CI)	*p* Value
**Age (years)**			
Children (0–12)	10/21(47.6)		
Teenagers (13–19)	1/15(6.7)		
Adolescents (20–23)	3/45(6.7)	0.986(0.946–1.027)	0.024
Adults (24–50)	8/32(25.0)		
Elders (51–Above)	5/29(17.2)		
**Gender**			
Female	20/96(20.8)	1.440(0.211–9.827)	0.001
Male	7/51(13.7)		
**Diagnosis**			
Urinary tract infections	4/36(11.1)		
Wounds *	6/28(21.4)		
Blood stream infection	2/11(18.2)	0.860(0.438–1.688)	0.002
High vaginal infection	2/25(8.0)		
Otorhea infections	13/35(37.1)		
Others **	0/8(0.0)		

*p* < 0.05 significant; * including skin and soft tissue infections, aspirate effusion, purulent pus; ** other infections including eye infection, throat, sputum and endocervical collections; *n*, number of MDR-Psa; N, total number of collected *Pseudomonas aeruginosa* strains; %, percentage rate.

**Table 2 antibiotics-12-00626-t002:** Detection rate of virulence factors produced by the MDR *Pseudomonas aeruginosa*.

Functional Activities		Otorrhea	Blood Stream Infection	UTI	Vaginal Infection	Wound Infection	95% CI	*p* Value
		*n*(%)						
Biofilm	Producer	13(48.2)	2(7.4)	3(11.1)	2(7.4)	5(18.5)	0.820–1.032	0.001
Efflux (EtBr (mg/L))	Efflux(0.5)	12(44.4)	2(7.4)	4(14.8)	1(3.7)	5(18.5)	0.762–1.016	0.001
	Efflux(1.0)	8(29.6)	2(7.4)	3(11.1)	1(3.7)	4(14.8)	0.209–0.606	0.001
	Efflux(1.5)	8(29.6)	2(7.4)	1(3.7)	0(0.0)	0(0.0)	0.477–0.857	0.001
	Efflux(2.0)	12(44.4)	2(7.4)	3(11.1)	1(3.7)	6(22.2)	0.762–1.016	0.001
Virulence factor	Protease	5(18.5)	1(3.7)	0(0.0)	1(3.7)	1(3.7)	0.112–0.480	0.003
	Lipase	11(40.7)	2(7.4)	2(7.4)	2(7.4)	3(11.1)	0.143–0.523	0.001
	Hemolysin	9(33.3)	2(7.4)	2(7.4)	1(3.7)	6(22.2)	1.109–1.780	0.001

*p* < 0.05 significant, UTI, urinary tract infection; *n*, number; %, percentage rate; EtBr conc, concentration (mg).

**Table 3 antibiotics-12-00626-t003:** Analysis of strain efflux, biofilm, lipase, protease and hemolysis production with MARI.

Parameters	Multi-Antibiotic Resistance Index (MARI)			
	R Coefficient	SE	95% CI	*p* Value
efflux	−0.85	0.04511	−0.1119–0.0743	0.6807
biofilm	−0.08	0.04424	−0.0226–0.1601	0.1334
lipase	0.36	0.09143	−0.0800–0.2982	0.2448
protease	0.28	0.1168	−0.2824–0.2009	0.7305
hemolysis	0.13	0.04520	−0.0673–0.1198	0.5669

## Data Availability

All data generated during the study are presented in this paper.

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
