# Peer review of "Evaluation of Efflux-Mediated Resistance and Biofilm formation in Virulent Pseudomonas aeruginosa Associated with Healthcare Infections"

_antibiotics, 2023, doi:10.3390/antibiotics12030626_

Round 1

Reviewer 1 Report

The manuscript from Akinduti et al evaluated the association of efflux-mediated resistance and biofilm formation in virulent Pseudomonas aeruginosa associated with healthcare infections in Nigeria. Firstly, the authors screened a collection of 147 strains for antibiotic resistance. Those strains with MDR phenotype were selected for the characterization of virulence factors.   The manuscript is interesting to the general audience of Antibiotics. However, the authors need to improve the organization and presentation of the manuscript.    The suggestions/questions are included in the attached pdf file.

Author Response

Reviewers’ Comments:
Reviewer 1

The manuscript from Akinduti et al evaluated the association of efflux-mediated resistance and biofilm formation in virulent Pseudomonas aeruginosa associated with healthcare infections in Nigeria. Firstly, the authors screened a collection of 147 strains for antibiotic resistance. Those strains with MDR phenotype were selected for the characterization of virulence factors.   The manuscript is interesting to the general audience of Antibiotics. However, the authors need to improve the organization and presentation of the manuscript.    The suggestions/questions are included in the attached pdf file.

Response: The organization and presentation of the manuscript is now improved.

Line 20-21: Please consider to rewrite this sentence to improve the readability and understanding.

Response: Revised appropriately

Line 43: correction

Response: it is corrected

Line 54: Please consider to rewrite this sentence to improve the readability and understanding

Response: Revised

The authors should make clearer the objective of the study. As the methods are presented after results and discussion, the authors should make a brief explanation about the sources and number of strains included in this study and the collection period.

Response: Objective is now revised to be more clearer. Details of the sources and number of strains included in this study and the collection period were listed in materials and methods section. However, brief information on the strain collection is now included in the result section.

It is confusing. The total number of MDR strains is 27 or 147? how the selection of this 27 strains was performed? The authors have to make this point clear for the readers

Response: The selection of the 27 MDR isolates were based on their resistance pattern from the antimicrobial susceptibility assay which showed 27 to be MDR. This is now well explained in Result section 2.1

Please provide a general caption for this figures

Response: General caption is now provided for Figure1 and 2, as indicated

Reviewer 2 Report

The manuscript describes the analysis of P. aeruginosa strains collected from patients in clinical settings. The work aimed to investigate the role of efflux systems and biofilm formation on the generation of antibiotic resistance. Surprisingly, the analysis showed a correlation between sex and antibiotic resistance, making doubt the original hypothesis.

The work is an essential surveillance study to analyze antibiotic resistance in hospital settings. There is no detailed analysis of the causes of observed findings except that females may have more recurrent infections.

Author Response

Reviewer 2

Comment 1: The manuscript describes the analysis of P. aeruginosa strains collected from patients in clinical settings. The work aimed to investigate the role of efflux systems and biofilm formation on the generation of antibiotic resistance. Surprisingly, the analysis showed a correlation between sex and antibiotic resistance, making doubt the original hypothesis.

Response: The analysis does not only showed the correlation between sex and antibiotic resistance but showed;

  1. Age–specificity, gender and infection sources as considerable risk factors for healthcare-associated multi-drug resistance (MDR) Pseudomonas aeruginosa infection facilitated by continuous declined demography posing a significant challenge to clinical management, hospital stay, and increase morbidity (Result section 2.3 and Table 2).
  2. The correlation of expressed virulence proteins (protease, lipase and hemolysin) and biofilm formation which enhance infection burden and therapeutic failure is demonstrated in Result section 2.4, Figure 2 and Table 3.; to buttress the hypothesis. Furthermore, the functional efflux and biofilm activities provide insight on the mechanism of P. aeruginosa-host adaptation in severe infection pathology which could aid understanding of the targets to be considered for new drug discovery.

Comment 2: The work is an essential surveillance study to analyze antibiotic resistance in hospital settings. There is no detailed analysis of the causes of observed findings except that females may have more recurrent infections.

Response: Detailed analysis of the antibiotic resistance in hospital settings and the causes (infection sources) were discussed in Table 2. Not only in females may recurrent infection occur but poor clinical outcomes in urinary and otorhea cases (Discussion section 3.0). The causes are already discussed in relation to the observed finings in first paragraph of the Discussion

‘Declining demography of the admitted adult group in these healthcare settings with poor socio-economic condition, prolonged hospital stay, and antibiotic misuse are common factors that persistently influence the infection spread [35]. Self-prescription and consumption of counter drugs are common practices that enhance MDR and continuous treatment failure’

Round 2

Reviewer 2 Report

The manuscript has been changed slightly. Mostly, the parts describing figures 1 and 2 and the paragraph in the introduction.

The work describes general findings related to antibiotic resistance in clinical isolates of P. aeruginosa and its distribution according to sex, place of isolation, and the type of antibiotics it is resistant to.

Data is presented clearly except for the legend of Fig. 2A/B, where the symbols of code are too small to tell the difference in many cases. It would be helpful to increase the size and improve the quality of pictures to at least 300dpi.

 A broader question remains regarding the meaning of the study. At this stage, there is a clear data correlation in Fig. 3 and 2. Table 3 is a separate issue, and it is clear the data do not correlate, possibly due to the complexity of the problems seen during human infection and the simplified way of presenting separate components as the virulence factors. In reality, the components work together, and the lack of clear distinction, as probably anticipated by the authors, is not surprising.

Overall, the study shows a clinical view of P. aeruginosa infections encountered in hospital settings in Nigeria and could have value in diagnosing and preventing future infections by correcting current practices, not only in hospitals but in patients.

Author Response

Reviewer 2

The manuscript has been changed slightly. Mostly, the parts describing figures 1 and 2 and the paragraph in the introduction.

The work describes general findings related to antibiotic resistance in clinical isolates of P. aeruginosa and its distribution according to sex, place of isolation, and the type of antibiotics it is resistant to.

Data is presented clearly except for the legend of Fig. 2A/B, where the symbols of code are too small to tell the difference in many cases. It would be helpful to increase the size and improve the quality of pictures to at least 300dpi.

 A broader question remains regarding the meaning of the study. At this stage, there is a clear data correlation in Fig. 3 and 2. Table 3 is a separate issue, and it is clear the data do not correlate, possibly due to the complexity of the problems seen during human infection and the simplified way of presenting separate components as the virulence factors. In reality, the components work together, and the lack of clear distinction, as probably anticipated by the authors, is not surprising.

Overall, the study shows a clinical view of P. aeruginosa infections encountered in hospital settings in Nigeria and could have value in diagnosing and preventing future infections by correcting current practices, not only in hospitals but in patients.

Response: Comments and observations provided are well appreciated and were already effected in the manuscript.  The symbols of the code in Fig. 2A/B is now enlarged and clearly presented in improved quality indicated by the Reviewer and it has helped to further improve the manuscript.